# The Synergistic Effect of Trace Ag and Hot Extruding on the Microstructure and Properties of a Biodegradable Mg-Zn-Sr-Ag Alloy

**DOI:** 10.3390/ma16196423

**Published:** 2023-09-27

**Authors:** Qifeng Shi, Huishu Wu, Zhixian Gao, Dongsheng Wang, Jingwen Wang, Youwen Yang, Runxia Li

**Affiliations:** 1College of Mechanical Engineering, Tongling University, Tongling 244000, China; 2New Copper-Based Material Industry Generic Technology Research Center of Anhui Province, Tongling 244000, China; 3Key Laboratory of Additive Manufacturing, Anhui Higher Education Institutes, Tongling University, Tongling 244000, China; 4Materials Science and Engineering, Dongguan University of Technology, Dongguan 523808, China; 5Institute of Additive Manufacturing, Jiangxi University of Science and Technology, Nanchang 330013, China; yangyouwen@jxust.edu.cn

**Keywords:** Mg-Zn-Sr-Ag alloy, microstructure, mechanical properties, corrosion resistance, extrusion deformation

## Abstract

To further improve the mechanical properties and corrosion resistance of the biodegradable magnesium (Mg) alloy, the Mg-4Zn-0.5Sr-xAg alloy (x = 0.2 wt.%, 0.5 wt.%, 1.0 wt.%, and 2.0 wt.%) was smelted in vacuum under the protection of inert gas. The effect of the Ag content on the microstructure and mechanical properties of Mg-4Zn-0.5Sr was tested. The results show that the comprehensive properties of Mg-4Zn-0.5Sr-0.5Ag are best. The grain size of the Mg-4Zn-0.5Sr-0.5Ag alloy is minimal, that is, 83.28 μm. The average tensile strength (σ_b_), yield strength (σ_s_), elongation (ε), and hardness for the Mg-4Zn-0.5Sr-0.5Ag alloy is 168.00 MPa, 88.00 MPa, 12.20%, and 59.90 HV, respectively. To further improve the properties of cast Mg-4Zn-0.5Sr-0.5Ag alloy, extruding treatment was conducted. After extrusion deformation, the grain size of the alloy was significantly refined to 9 μm; at the same time, fine second phases were formed and evenly distributed in the matrix. And then, the mechanical properties of the alloy are significantly enhanced due to the effect of fine crystal strengthening and dispersion strengthening. The σ_b_, σ_s_, ε, and hardness value for the extruded Mg-4Zn-0.5Sr-0.5Ag alloy are 236.00 MPa, 212.00 MPa, 18.97%, and 65.42 HV, respectively. Under the synergistic action of adding the Ag element and extrusion treatment, the grain size of the alloy was significantly refined and the coarse second phase in the alloy became refined to disperse in the matrix, which benefits the formation of electric couples characterized as small cathode–large anode between the second phase and Mg matrix. During full immersion, corrosion products covered on the large anode surface could reduce the galvanic corrosion tendency.

## 1. Introduction

With the development of medical metal materials, biodegradable magnesium (Mg) and its alloys have aroused the keen interest of researchers [1,2,3]. Especially when applied as a biomedical orthopedic, Mg is a promising alternative to traditional stainless-steel nails and titanium nails [4]. This is primarily due to its ability to degrade and be absorbed by the body safely under physiological conditions, which allows patients to avoid the pain caused by the surgery for implant removal, and solves the shortcomings of implant removal in clinical application [5]. Another vital superiority of the Mg alloy is that its modulus is similar to bone, which prevents the stress shielding of bone tissue effectively [6]. Pure magnesium has the best biocompatibility, but poor strength and hardness; therefore, the mechanical integrity of pure Mg is enormously easy to destroy before the tissue has sufficiently healed, which limits the application of pure Mg. Due to the function of alloying, the mechanical properties of Mg alloys are better than those of pure Mg, so research about Mg alloys applied in the biomedical field has dramatically increased.

At the outset, off-the-peg Mg alloys, such as AM60B and WE3, have been attempted to be used as biomaterials [7,8]. However, these industrial alloys turned out to be unfriendly to humans; this is mainly related to the composition of the strengthening elements in these magnesium alloys, such as manganese (Mn), aluminum (Al), zirconium (Zr), and rare earth elements [9,10,11]. For example, the high concentration of Mn has been confirmed to cause neuro-toxicity and lead to Parkinson’s disease [12]; an excess of ionic Al would induce dementia [13]; and a large amount of rare earth elements such as Zr and Y has been tested to be associated with liver, lung, breast, and nasopharyngeal cancers [14]. Moreover, Mg alloy used as a degradable material is, mainly, by its own characteristic, easily corrodible. However, higher requirements for the corrosion properties of magnesium alloys are also needed to be put forward. Firstly, the hydrogen produced by the fast degradation of the magnesium alloys is released too quickly to be absorbed by the human body, which would cause severe tissue necrosis. Secondly, local corrosion was prone to take on magnesium alloys; there is a high probability this would lead to premature or sudden material failure [15,16]. Surface technologies of coating, electroless deposition, physical vapor deposition, etc. have been used to delay the degradation process of the magnesium alloy [17,18]. Nevertheless, if the inherent corrosion resistance of the magnesium alloy is insufficient, the alloy would still be corroded rapidly when the protective layer of the alloy is damaged. Therefore, it is urgent that we develop human-friendly Mg-based biomaterials with a strong inherent corrosion resistance to meet clinical application needs. 

Alloying is an effective method for improving the intrinsic property of magnesium alloys [19]. Mg-Zn alloys are common binary medical alloys. Research has proven that Mg-Zn alloys exhibit good cytocompatibility. At first, the cytocompatibility of Zn is related to the fact that Zn is an essential element for the human body and it is critical to its biological function. In addition, Zn and other impurity elements will generate corrosion products and cover the alloy surface when immersed in body fluid, which can effectively block the permeation of the corrosive medium and avoid direct contact between corrosive medium and the matrix, thereby reducing the degradation rate of the alloy [20]. The mechanical property of Mg-Zn alloy is owed to the solid solution property of Zn in Mg. The maximum solid solubility of Zn in Mg is 6.2 wt.%. When element Zn was added to Mg, the dual effect of solution strengthening and aging strengthening is aroused [21]. Hence, Mg–Zn series is an important Mg-based alloys developed for structural materials.

However, the mechanical property for simple Mg-Zn binary alloys is moderate and that is insufficient to meet the requirement of a stent. Therefore, some ternary and quaternary alloys are continuously developed, such as Mg-Y, Mg-Zn-Zr, Mg-Zn-Gd, Mg–Nd–Zn–Zr, etc. [22,23,24,25]. Strontium (Sr) is an important additive in Mg-based alloys for improving relatively high-temperature mechanical properties. Moreover, the element Sr has many medical advantages; for example, the appropriate amount of the Sr element in the body can prevent the formation of arteriosclerosis and thrombosis [10]. In N. Birbilis’ work, the mechanical and degradation properties of the Mg-Sr and Mg-Zn-Sr alloys were investigated [26]. The results indicate a decrease in grain size and an increase in strength with increasing Sr and Zn content. However, excessive Sr elements in the Mg alloy will facilitate the formation of brittle phases and were assembled at the grain boundaries, which would be bound to lower the mechanical property and corrosion performance. Therefore, Zhen Li et al. explored the effect of trace Sr on the mechanical properties of the Mg-Zn alloy. However, adding only trace Sr has no apparent elevation effect on the mechanical property, and, when trace Zr was added into the Mg-Zn-Sr alloy, the mechanical properties of the Mg-Zn-Sr-Zr alloy have significantly improved through the synergistic effect of Zr and Sr [11]. In addition, Ag is a kind of antibacterial element, which has been applied in clinical medicine [27]. When used as implants, the quantity of platelets adhered on the Mg-Ag alloy dies off rapidly compared to pure magnesium [28,29]. Moreover, studies have shown that adding the appropriate amount of the Ag element to the Mg alloy can effectively reduce the grain size and improve the mechanical properties of the alloy due to the large solid solubility of Ag in magnesium alloy (which is 15.5 wt.%) [30,31]. Hence, Ag is a promising additive with which to optimize the properties of magnesium alloys.

Apart from the composition, the mechanical and degradation properties also have a strong dependence on the microstructure [32]. Many experiments demonstrated that post-extrusion treatment can refine the grain of the alloy and homogenize the second phase in the casted alloy, which will significantly enhance the strength of the alloy and give the alloy a stable degradation rate. For example, Peng and Tie et al. used backward extrusion to optimize the microstructure of the Mg-Zn-Sr alloy, which refines the average grain size of the alloy and makes the secondary precipitates in the matrix evenly distributed [33,34]. In the work of ZG Xu et al., hot rolling was used to improve the mechanical properties of the Mg alloy [35]. Therefore, it is an effective means with which to improve the comprehensive properties of the alloy by adding the appropriate elements and compounding a variety of processing techniques.

In our previous work, Mg-4.0 wt.%Zn-0.5 wt.%Sr was best optimized with both mechanical and degradation. Mg-Zn alloys are promising to be developed as ideal biodegradable implants for bone tissue engineering. The Sr element is an important component of human bone. In our previous work [36], Sr was added to improve the biocorrosion and mechanical properties of the Mg-4Zn alloy; 0.5 wt.%Sr is the most appropriate content for corrosion and mechanical properties. However, the property still has room for improvement. The element Ag is a good grain refiner and kind to the human body, by which the novel casted Mg-4Zn-0.5Sr-xAg alloys were developed and studied in this work. Considering the deficiency of the casting process, the Mg-4Zn-0.5Sr-xAg alloy is further subjected to hot extrusion treatment. The synergic effects of trace Ag and extrusion treatment on the microstructure and mechanical and degradation properties of the extruded Mg-Zn-0.5Sr-xAg alloy for biomedical applications are presented.

## 2. Experimental

### 2.1. Sample Preparation

In this work, commercially pure magnesium ingot (Mg), Zinc ingot (Zn), silver wire (Ag), and the Mg-Sr master alloy (with 20 wt.%Sr) were mixed to smelting Mg-4Zn-0.5Sr-xAg alloys (x = 0, 0.2, 0.5, 1.0, and 2.0 mass%). Detailed steps are as following: Mg ingot were melted in electric resistance furnace at 780 °C; protective gas of SF_6_ and N_2_ with a volume ratio of 2:98 was introduced when the furnace temperature rose to 600 °C. Then, silver wire was added to melted Mg ingot and stirred for 2 min. When the melt cooled to 720 °C, Mg-Sr master alloy and Zn ingot were added in sequence, and stirred for 4 min and 5 min, respectively. To realize the homogenization of alloying elements and settlement of inclusions, the molten alloy was held for 30 min under gas protection. Finally, molten alloy was poured into a ceramic mold at 720 °C, by which the Mg-5Zn-0.5Sr-xAg alloys were obtained. Particle compositions (Par.) of the casted alloys were detected by spark direct reading spectrometer and listed in Table 1. The theoretical composition (Theo.) for the alloys are also listed for comparison. In this work, the ceramic mold for molten metal was a cylindrical ceramic mold: the outer diameter was 115 mm, the inner diameter was 90 mm, and the inner wall of the mold was brushed with ZnO paint. The Mg-4Zn-0.5Sr-xAg ingots were machined into cylinders with a diameter of 20 mm and length of 65 mm.

### 2.2. Hot Extrusion Treatment

To further improve the mechanical properties, the cast Mg-5Zn-0.5Sr-xAg alloy was treated by hot extrusion. Before extruding, the cast Mg-5Zn-0.5Sr-xAg alloy was turned into a cylinder sample with diameter of Φ20 mm and homogenized by placing it in heat treatment furnace for 12 h at 440 °C. Then, the cylinder sample was extruded into bars with a diameter of Φ8 mm using an YH61-500G extruder at 400 °C with an extrusion ratio of 6.25. To prevent cracks when magnesium alloy contacted with cold mold, the cast alloy and mold should be preheated at 400 °C for 2 h before extruding. It should be noted that the cast Mg-5Zn-0.5Sr-xAg alloy with the best Ag content is selected for extrusion treatment. 

### 2.3. Charaterization

The microstructure and the grain size of the cast and extruded Mg-4Zn-0.5Sr-xAg (x = 0~2.0 wt.%) alloys were examined using optical microscopy (OLYMPUS GX51, Aierfa, Changzhou, China), scanning electron microscopy (SEM), and transmission electron microscopy (TEM), which were used to observe the second phase of the alloys, and the EBSD test of the alloy samples was carried out by the electron backscatter diffract graph equipped with the TSL. The phase components of Mg-4Zn-0.5Sr-xAg alloys were detected by X-ray diffraction (XRD, Rigaku D/max/2500 PC, Rigaku, Tokyo, Japan) with CuKa radiation and a scanning speed of 8°/min. The XRD pattern analysis was performed using the “MDI jade 5.0” software 2019. The XRD patterns were compared with standards compiled by the JCPDS standards. For optical micromorphology, all samples were ground by silicon carbide (SiC) water-abrasive paper gradually from 600# to 3000#, and followed by a Al_2_O_3_ polishing process (0.3 μm), after which the polished surface was etched with 5% nitrate alcohol solution for 8~10 s. Sample surface for EBSD test was treated as follows: Firstly, the alloy samples were pre-ground and mechanical polished, and then the surface of the mechanical polished samples was electrolyzed. The electrolyte was a mixture of 10% perchloric acid and 90% anaqueous ethanol, and liquid nitrogen was introduced into the electrolyte to make the electrolytic environment temperature between −35 °C and −30 °C. During the electrolytic process, the current was 0.15 mA, the voltage was 15 V, and the electrolytic time was 85 s.

### 2.4. Properties

#### 2.4.1. Mechanical Properties

Hardness test was conducted on Vickers hardness tester with the load of 0.98 N and a holding time of 30 s. Tensile tests conducted on electronic universal testing machine (WGW-100H) was used to determine the yield strength (YS), ultimate tensile strength (UTS), and elongation of the alloys. Tensile specimens were machined to rectangular dog-bone-shaped samples following the specifications of GBT228-2002. The tensile test was performed in air at room temperature with a tensile speed of 0.3 mm/min. Tests on three duplicate samples have been performed to evaluate the mechanical properties of the alloys.

#### 2.4.2. Degradation Properties

The degradation rate was tested in Hanks’ balanced salt solution (HBSS) for 15 days with an immersion ratio of 0.03 cm^2^/mL (the ratio of surface area to solution volume). The tested samples were cut into pieces with dimensions of 6 mm × 10 mm. During immersion, the pH value of the solution was measured with a digital pH meter (STARTER 3100, OHOUS) every 2 h. Mg-based alloys which were immersed in HBSS would dissolve, as in the following chemical equation:Mg + 2H_2_O → Mg^2+^ + 2OH^−^ + H_2_ ↑(1)
by which the corrosion rate was calculated by measuring the weight loss before and after immersing, and corrosion rate were calculated by using the following equation:C_R_ = *K*(W_0_ − W_1_)/(ADT)(2)
where C_R_ is the corrosion rate (mm/y), W_0_ and W_1_ are the weights of sample before and after immersion, A is the surface area (cm^2^), T is the immersion time, and D is the density (g/cm^3^) of the material. The test sample before immersing was sanded, polished, and rinsed, in that order, by ethanol and acetone, and blow-dried by the end. The weight and surface area of the dried sample were measured and recorded as W_0_ and A. W_1_ was measured after cleaning and removal of all corrosion products in chromic acid (180 g/L, 20 min immersion at room temperature). The weight loss was measured after immersing for 2 d, 4 d, and 8 d to calculate the corrosion rate. The morphology of samples after immersion for 2, 4, and 8 days was also analyzed by SEM. Before SEM analyzing, the samples were gently rinsed with absolute ethyl alcohol and dried at room temperature.

## 3. Results and Discussion

### 3.1. The Microstruture and Mechanical Properties of Casted Mg-Zn-Sr-xAg

Microstructure and Composition of Casted Mg-Zn-Sr-xAg

Figure 1 presents the metallographic structure of Mg-4Zn-0.5Sr alloys with different Ag contents; the grain size of the alloys was computed by the Ipwin32 software 2018, as shown in Table 2. The alloys are mainly composed of the grain boundaries and matrix. From the metallographic structure of casted Mg-4Zn-0.5Sr alloy, the average grain size for the Mg-4Zn-0.5Sr alloy is 103.62 μm, and the second phase particles (which are punctiform) are distributed along the grain boundary and inside the grain. The grain size for the Mg-4Zn-0.5Sr-0.2Ag alloy is 102.67 μm, and the amount of second phase particles at the grain boundary is increased compared with that of Mg-4Zn-0.5Sr alloy. When the Ag content in the Mg-4Zn-0.5Sr alloy is 0.5 wt.%, the crystal particle is refined to 83.28 μm and more well-distributed. At the same time, the punctiform second phases distributed in the matrix were significantly reduced or had evenly disappeared. 

Ag has a large solid solution degree and vacancy binding energy in magnesium alloy, which can not only play a role in solid solution strengthening in the matrix, but also effectively prevent the growth of the precipitated phase in the aging process [12]. However, the precipitates at the grain boundaries gradually increased and showed a tendency of segregation with the addition of Ag. When the Ag content was 1.0 wt.%, the precipitates in the Mg-4Zn-0.5Sr-1.0Ag alloy were more biased; at the same time, the average grain size of the alloy is increased to 102.32 μm. And when the Ag content was up to 2.0 wt.%, the grain size of the alloy enlarges to 110.43 μm, which is not beneficial for the mechanical property. Therefore, the best addition amount of Ag content in the alloy is 0.5 wt.%.

Figure 2a shows the XRD patterns of Mg-4Zn-0.5Sr-xAg. The results show that, for all the samples, the major phases are the magnesium (α-Mg) matrix [33], MgZn, and Mg_17_Sr phase [37]. For the casted Mg-4Zn-0.5Sr-xAg alloy, the peaks from the Ag-containing phases are weak, which may be due to some Ag atoms being retained in the solid-solution state, or forming precipitates that are too little to be resolved by the XRD experiment [38]. With the Ag content goes up to 2%, the diffraction peak intensity of Mg_4_Ag at 28° is increased. At the same time, the intensity change and shift in position of the peaks assigned to MgZn and Mg_17_Sr_2_ were observed, which is related to the variation in the lattice constant of α-Mg caused by the adding of Ag. Then, different degrees of solid-solution strengthening effects can result. As shown in the Figure 2b, the intensity of the peaks assigned to MgZn (37°) and Mg17Sr2 (34°) is decreased after extruding treatment. This is because the second phase is dissolved into the matrix during the homogenization process of hot extrusion treatment, and the amount of precipitation is greatly reduced after extrusion (as shown in SEM image of extruded Mg-4Zn-0.5Sr-0.5Ag alloy in Figure 8), so the diffraction peak intensity of the second phase is significantly reduced.

The microstructure of the Mg-4Zn-0.5Sr-xAg alloy was further analyzed by SEM as shown in Figure 3. The detailed element composition of precipitates in Mg-4Zn-0.5Sr and Mg-4Zn-0.5Sr-0.5Ag was investigated by EDS and listed in Table 3. The alloys are mainly composed of the gray matrix phase and precipitated second phase. The morphology of the precipitated phase is mainly a point-like phase (as shown at point A D, and E in Figure 3), irregular round-like phase (as shown at point B in Figure 3), and long strip phase (as shown at point C and F in Figure 3). As demonstrated in the EDS results in Table 2, the second phase “A” and “B” mainly consist of Mg and Zn, and the irregular round-like second phase “B” contains bits of Sr and Ag. Compared with the irregular second phase “B”, the Sr content in the long strip shape (C, E, and F) significantly increased. When we combine the phase analysis, it can infer that the gray matrix is mainly α-Mg, and the bright white precipitated phase is MgZn, Mg_17_Sr_2_, and a small amount of Mg_4_Ag. According to the EDS results, it is safe to deem that the irregular phase is mainly made up of the MgZn phase, and the long strip phase consists of Mg_17_Sr_2_ and a small amount of Mg_4_Ag.

2.Mechanical Properties of Mg-4Zn-0.5Sr-xAg

The tensile strength (σ_b_), yield strength (σ_s_), and elongation (ε) for the casted Mg-4Zn-0.5Sr-xAg were tested by tensile testing, as shown in Figure 4, and the hardness of the alloys was also tested. The results were presented in Table 4. For the Mg-4Zn-0.5Sr alloy, the σ_b_, σ_s_, and ε value are 161.00 MPa, 82.00 MPa, and 10.50%, respectively. With the increase of Ag content, the tensile strength, yield strength, and elongation of the alloy increased first and then decreased. When the Ag content was 0.5 wt.%, the mechanical properties of the alloy were the best. The σ_b,_ σ_s_, and ε value for Mg-4Zn-0.5Sr-0.5Ag are 182.00 MPa, 102.00 MPa, and 13.81%, which are enhanced by 13%, 24.4%, and 31.5% compared with that of Mg-4Zn-0.5Sr. When the Ag content in Mg-4Zn-0.5Sr is up to 1 wt.%, the mechanical properties slightly decreased in comparison to Mg-4Zn-0.5Sr-0.5Ag, but are still superior to Mg-4Zn-0.5Sr. The σ_b_, σ_s_, and ε value of Mg-4Zn-0.5Sr-1Ag are 175.00 MPa, 85.00 MPa, and 13.12%, respectively. As for Mg-4Zn-0.5Sr-2Ag, the mechanical properties have apparently fallen. The σ_b_, σ_s_, and ε value for the Mg-4Zn-0.5Sr-2Ag alloy are 129.00 MPa, 70.50 MPa, and 6.67%, respectively, which have dropped by 20%, 14%, and 35% compared with the values of Mg-4Zn-0.5Sr. The hardness of the alloys was increased in this order—Mg-4Zn-0.5Sr, Mg-4Zn-0.5Sr-1Ag, Mg-4Zn-0.5Sr-0.2Ag, and Mg-4Zn-0.5Sr-0.5Ag, with values of 43.15HV, 51.6 HV, 52.9 HV, and 59.90HV, respectively. Among them, the hardness for Mg-4Zn-0.5Sr-0.5Ag is the highest.

The tensile test showed that the as-cast Mg-4Zn-0.5Sr-0.5Ag alloy has the best mechanical properties. In order to investigate the mechanical property thoroughly, the morphology of the tensile fracture (Figure 5) and near the fracture (Figure 6) were observed by SEM. In fracture analysis, dimples and tearing edges are often used to judge the plastic property. During the fracture process, the more dimples and tearing edges on the micro-structure, the more energy is required during fracture, indicating a better plasticity of the alloy [10].

As shown in Figure 5, the tensile fracture of all alloys displayed three characteristic morphologies—tearing edge (laceration), dissociation platform, and dimple—which indicates that the deformation of Mg-4Zn-0.5Sr-xAg is mainly plastic deformation. For Mg-4Zn-0.5Sr, obvious cleavage platforms were detected on the fracture surface. For Mg-4Zn-0.5Sr-0.2Ag, dimples were gradually generated on the fracture surface, and the cleavage platform gradually degenerated without obvious tearing edges, which implied that the addition of Ag could enhance the plasticity of Mg-4Zn-0.5Sr. When the Ag content in Mg-4Zn-0.5Sr is 0.5 wt.%, a certain number of dimples and cleavage were detected on the fracture surface, indicating that alloy is fractured by the ductile and brittle method. When the Ag content incremented continually, the quantity of dimples declined and a large number of cleavage facets appeared. When the Ag content in Mg-4Zn-0.5Sr reached 2.0%, river-like tearing edges were focused on the fracture of Mg-4Zn-0.5Sr-2.0Ag, which resulted in a large area of cleavage fracture and a significantly decrease in plasticity. Combined with the tensile test results, it can be assumed that the Mg-4Zn-0.5Sr-0.5Ag alloy is in possession of the best plasticity among the Mg-4Zn-0.5Sr-xAg alloys.

The microstructure near the fracture of the cast Mg-4Zn-0.5Sr-xAg (x = 0~2.0 wt.%) alloys was exhibited in Figure 6. As demonstrated, some pits appeared on the verge of the fracture notch of the Mg-4Zn-0.5Sr alloys and there was no obvious second phase and crack, which indicates that the deformation of the Mg-4Zn-0.5Sr alloy is mainly brittle deformation. Compared with Mg-4Zn-0.5Sr, the morphology near the fracture of Mg-4Zn-0.5Sr-0.2Ag is quite different. Firstly, some tiny second phase particles were detected. Secondly, small and medium-sized pores gather into large pores during the deformation process, which become the primary crack source. Meanwhile, along the tensile direction, microcracks were generated in large quantities around the pores, which may be the incubation period of the fracture. From the morphology near the fracture of Mg-4Zn-0.5Sr-0.5Ag, a clear slip band appeared near the fracture, and the second phase particles are pinned to the slip zone, which strengthened the plastic deformation capacity of the Mg-4Zn-0.5Sr-0.5Ag alloy. When the Ag content in the alloy further increased, slip bands near the fracture were scattered and the quantity of the slip bands was cut down visibly. When the Ag content had gone up to 2%, cracks interacted and converged to form larger cracks, thereby resulting in a brittle fracture and sharp decline in mechanical properties.

### 3.2. The Microstructure and Mechanical Properties of Extruded Mg-Zn-Sr-0.5Ag

From the above analysis, it can be concluded that 0.5 wt.% Ag in Mg-4Zn-0.5Sr alloy can improve the mechanical properties of the alloy, but, due to the segregation of the needle-like second phase, the mechanical properties of the alloy were not significantly improved. In order to further improve the mechanical properties of the magnesium alloy, the as-cast Mg-4Zn-0.5sr-0.5Ag alloy is treated by hot extrusion. The microstructure and mechanical properties of the extruded samples were carried out.

Figure 7 and Figure 8 show the metallographic morphology and SEM morphology for the Mg-4Zn-0.5Sr and Mg-4Zn-0.5Sr-0.5Ag alloy before and after extrusion treatment. As presented in the metallograph, the grain size of the alloy after being extruded was significantly reduced. The average grain size of the extruded Mg-4Zn-0.5Sr alloy is 20 μm, which is reduced by 81% compared with that of the cast Mg-4Zn-0.5Sr alloy, and the grain size for the extruded Mg-4Zn-0.5Sr-0.5Ag is 9 μm, which is reduced by 89% compared with that of the cast Mg-4Zn-0.5Sr-0.5Ag alloy. From the SEM images, it can be noticed that the long and coarse second phases of Mg_17_Sr_2_ were mainly distributed on the grain boundary before extrusion treatment. After being extruded, the coarse second phases in the alloy were broken up into fine second phases and distributed in the matrix evenly. The addition of the Ag element increased the quantity of the second phase in the matrix, which was conducive to dispersion strengthening. In order to investigate the effect of extrusion deformation on the second phase of the alloy, the X-ray diffraction analysis of the extruded Mg-4Zn-0.5Sr-0.5Ag alloy was carried out, as shown in Figure 2b. From Figure 2a, it can be observed that the diffraction peak intensity of the second phase in the extruded Mg-4Zn-0.5Sr-0.5Ag alloy is higher than that in the cast Mg-4Zn-0.5Sr-0.5Ag alloy. The extrusion process of Mg-4Zn-0.5Sr-0.5Ag involves the two processes of homogenization and hot extrusion. Homogenization makes the second phase of the alloy dissolve into the matrix. After that, the dissolved second phase breaks up to form small second phase precipitation by extrusion treatment, which increases the concentration of the second phase and, thereby, results in a higher intensity of the diffraction peak, but there was no new second phase formed.

Further, the mechanical properties of the extruded alloys were measured by a tensile test, and the results were pictured in Figure 9. As depicted, extrusion treatment and adding the Ag element can effectively improve the tensile properties of the cast alloy. Firstly, the average σ_b_, σ_s_, and λ value for the extruded Mg-4Zn-0.5Sr alloy were 218.67 MPa, 184.00 MPa, and 15.93%, respectively, which have increased by 35%, 124%, and 54.7% compared with the values of the cast Mg-4Zn-0.5Sr alloy. Further, the average σ_b_, σ_s_, and ε value for the extruded Mg-4Zn-0.5Sr-0.5Ag alloy increased by 44.5%, 107.8%, and 36.9%, compared with the as-cast Mg-4Zn-0.5Sr-0.5Ag alloy, which are 263.00 MPa, 212.00 MPa, and 18.97%, respectively. Moreover, the average σ_b_, σ_s_, and λ value for the extruded Mg-4Zn-0.5Sr-0.5Ag alloy increased by 63%, 158.5%, and 83.5%, respectively, compared with the as-cast Mg-4Zn-0.5Sr alloy. Therefore, it can be concluded that the properties of the Mg-4Zn-0.5Sr alloy were remarkably strengthened under the synergistic action of the Ag addition and extruding treatment.

Figure 10 shows the tensile fracture morphology of extruded alloys; the tensile fracture of alloys were composed of a dimple and fine cleavage plane. A through tearing ridge appeared on the fracture of the extruded Mg-4Zn-0.5Sr alloy; therefore, the Mg-4Zn-0.5Sr alloy only handled with extruding treatment still fractured, as a typical brittle fracture. After adding Ag, the fracture surface of the extruded Mg-4Zn-0.5Sr-0.5Ag alloy is mainly composed of a dimple and the cleavage plane quantity, which indicates that the fracture type for the extruded Mg-4Zn-0.5Sr-0.5Ag alloy is a typical ductile fracture. From the above analysis, it is safe to deem that the mechanical properties of the cast Mg-Zn-Sr alloy can be significantly improved by the synergy of the addition of a trace Ag element and extruding treatment.

### 3.3. Corrosion Property of Extruded Mg-4Zn-0.5Sr-0.5Ag Alloy

In addition to the excellent mechanical properties, biomedical magnesium alloys also have certain requirements for the degradation performance, and their degradation characteristics are mainly related to the long-term service stability in body fluid. Therefore, the corrosion properties of the cast Mg-Zn-Sr-0.5Ag and extruded Mg-Zn-Sr-0.5Ag alloys were tested by full immersion in simulated body fluid (SBF). The pH value of the solution after different soaking times was measured. The corrosion rate of the alloy during different soaking times (after 2 d, 4 d, and 8 d) was test by the weight-loss method, and the corresponding corrosion surface morphology were detected.

The pH value variation tendency of SBF after being immersed with different alloys for different times was depicted in Figure 11. The variation tendency of the pH value for the SBF immersed with different alloys is similar. At first, the value increased sharply at the incipient immersion time (before 8 h). And then, the pH value of the solution sped up with the increase of immersion time, and gradually stabilized in the later period of immersing. Moreover, the rising slope for the pH value of the SBF immersed with alloys at the initial immersion time is increased in the order of the cast Mg-4Zn-0.5Sr, cast Mg-4Zn-0.5Sr-0.5Ag, extruded Mg-4Zn-0.5Sr, and extruded Mg-4Zn-0.5Sr-0.5Ag alloy, which indicates that the extruded Mg-4Zn-0.5Sr-0.5Ag alloy has the strongest activity. The higher activity of the Mg alloy would accelerate the dissolution of Mg when it was immersed in solution, which caused the aggregation of OH^−^, and, thereby, the rapid rise of the pH value. Prolonging the immersing time, the pH value of the SBF immersed with the cast Mg-4Zn-0.5Sr alloy was elevated continually and up to 10.51 after 160 h immersion, which suggested that the cast alloy was degraded persistently during the immersion period. However, the pH value of the SBF immersed with the extruded Mg alloy elevated gently with the prolonging of the immersion time. The pH value for the SBF after immersing with the extruded Mg-4Zn-0.5Sr-0.5Ag alloy was stabilized at 9.21, which may be the result of the corrosion products generated on the alloy surface that prevented the degradation of the alloy. Hence, the activity of the Mg alloy could promote the generation of a protective corrosion product layer, which would prevent the permeation of the corrosive medium and provide benefits for the long-term service stability of the alloy.

Table 5 shows the corrosion weight loss rate (C_R_) for the cast Mg-4Zn-0.5Sr and Mg-4Zn-0.5Sr-0.5Ag, and extruded Mg-4Zn-0.5Sr and Mg-4Zn-0.5Sr-0.5Ag alloys tested after 2 d, 4 d, and 8 d of immersion. The C_R_ value is calculated according to Formula (2). As represented in Table 5, the C_R_ value for the casted alloy is higher than that of the extruded alloy, and the addition of Ag could decrease the C_R_ value. At the same time, the C_R_ value for the alloys decreased with the prolonging of the immersion time, and the decline rate of the C_R_ for each alloy is different.

For the casted Mg-4Zn-0.5Sr alloy, the C_R_ is 1.05 mg/cm^2^/day after 2 d of immersion. The addition of Ag could decline the C_R_ of the cast Mg-4Zn-0.5Sr-0.5Ag alloy and the C_R_ value for the alloy is 0.90 mg/cm^2^/day after 2 d of immersion. The simple extrusion treatment has a weak influence on reducing the C_R_ value of alloy, and that value for the extruded Mg-4Zn-0.5Sr is 0.95 mg/cm^2^/day. However, under the synergistic effect of the Ag elements and hot extrusion treatment, the C_R_ value for the extruded Mg-4Zn-0.5Ag-0.5Sr-0.5Ag markedly decreased to 0.44 mg/cm^2^/day, which was decreased by 50% compared with that of the cast Mg-4Zn-0.5Sr alloy. With immersing, the C_R_ value for the alloys presented a decreasing trend. Compared with the C_R_ value measured at 2 d of immersion, that for the cast Mg-4Zn-0.5Sr alloy declined by 18% and 32% after 4 d and 8 d of immersion, which was 0.86 mg/cm^2^/day and 0.71 mg/cm^2^/day, respectively. For the cast Mg-4Zn-0.5Sr-0.5Ag, the C_R_ value at 4 and 8 d of immersion is 0.68 mg/cm^2^/day and 0.54 mg/cm^2^/day, which has declined by 24% and 40% compared with the values measured at 2 d of immersion. The C_R_ value for the extruded Mg-4Zn-0.5Sr-0.5Ag is the lowest under the synergy of the Ag addition and extruding treatment, which is 0.34 mg/cm^2^/day at 4 d of immersion and 0.25 mg/cm^2^/day at 8 d of immersion, respectively.

From the C_R_ value variation trend of alloys, it can be concluded that: 1. The addition of a trace Ag element and extrusion treatment can improve the corrosion resistance of the Mg-4Zn-0.5Sr alloy, and the synergy of the Ag element and extrusion treatment can significantly improve the corrosion resistance of the alloy; 2. Prolonging the immersion time, the C_R_ value of the alloys all gradually decreased, which may be related to the shielding property of the corrosion products formed on the alloy surface. Among them, the C_R_ value for the extruded Mg-4Zn-0.5Sr-0.5Ag alloy decreased most obviously during immersion, which suggests that the corrosion products on the extruded Mg-4Zn-0.5Sr-0.5Ag alloy have best corrosion resistance.

Figure 12 shows the etching surface of alloys that can be wiped off the corrosion products after immersing in SBF for different times. After 2 d of immersion, the corrosion of the cast Mg-4Zn-0.5Sr alloy mainly derived from the grain boundary; numerous corrosion gullies were generated and a lot of pitting corrosion also formed. When the alloy was immersed for 4 days, the corrosion spread along the grain boundaries, which would widen the corrosion gully. At the same time, the quantity and size of the pitting on the Mg-4Zn-0.5Sr surface were enlarged. After 8 d of immersion, the corrosion gullies gradually grew to form large corrosion pits, and some corrosion pits grew deep into the crystal, which would result in serious corrosion destruction. As for the cast Mg-4Zn-0.5Sr-0.5Ag, a light etching occurred on the surface of the alloy when immersed for 2 d. After 4 d of immersion, a large area of corrosion was formed and the etched depth gradually deepened after 8 d of immersion. The type of etching for the extruded Mg-4Zn-0.5Sr alloy is mainly pitting corrosion at the initial immersion time; the formed corrosion pits on the surface were fine and distributed evenly. The extruded Mg-4Zn-0.5Sr-0.5Ag alloy has a good corrosion stability under the synergistic effect of the trace Ag and extrusion process. At the beginning of immersion, a weak corrosion happened on the surface of the alloy. At 4 d of immersion, the corrosion area on the alloy surface was increased. With immersion, the surface presented a uniform corrosion with a river pattern, which is a typical form of general corrosion, and that is the main reason for the reduction of the C_R_ value of the extruded Mg-4Zn-0.5Sr-0.5Ag alloy.

### 3.4. Discussion

Based on the analysis above, it can be concluded that: under the synergistic effect of the Ag element and extrusion process, the mechanical properties of the extruded Mg-4Zn-0.5Sr-0.5Ag alloy are significantly enhanced. At the same time, the activity of the alloy is enhanced, which is beneficial in improving the stability of the alloy during long-term immersion. The mechanical properties of the extruding Mg-4Zn-0.5Sr-0.5Ag alloy are significantly enhanced, which can be attributed to the effect of fine crystal strengthening and dispersion strengthening.

Firstly, it is the function of fine crystal strengthening. According to the formula of Hall–Petch, σs = σ0 + kd^−1/2^, it can be inferred that the smaller the grain size of the alloy, the higher the strength. The refinement of the alloy grains increases the amount of grains in the alloy. If an external force were to be applied on the alloy, the stress concentration on each grain will be divided, by which the co-ordinated deformation ability of the alloy would be enhanced; meanwhile, the plasticity of the alloy would become excellent. On the other hand, the presence of refined grains means a large quantity of the grain boundaries. When the alloy suffered from external force, the dislocation migration will be hindered by grain boundaries, which would result in the plugging of dislocation, which would then improve the strength significantly. Based on the results of the metallographic analysis (Figure 7), the grain size of the extruded Mg-4Zn-0.5Sr-0.5Ag alloy was refined. Figure 13 is the electron backscatter diffraction (EBSD) of the extruded Mg-4Zn-0.5Sr-0.5Ag alloy. From the EBSD results, it can be seen that the grain size of the alloy is lower than 40 μm and the average size for the alloy is 10 μm. The grain refinement is mainly related to the solid solution of Ag in the Mg-4Zn-0.5Sr alloy. The maximum solid solution of Ag in the Mg alloy is 15.5 wt.%; a location will be vacated to a form vacancy when trace Ag was added in the Mg alloy. The vacancy binding energy can inhibit the growth of grains and refine grain.

Dispersion strengthening is related to the formation of the second phase. According to the SEM images, an increase in the number of second phase particles in the alloy with the Ag addition is observable. However, excessive Ag in the alloy not only enlarges the grain size, but also causes a large amount of precipitation. As shown in Figure 3, the long strip of Mg_17_Sr_2_ gradually segregated towards the grain boundary to form a coarse grain boundary, which results in a significant decline in the mechanical properties. Among the cast alloys, the mechanical property for the cast Mg-4Zn-0.5Sr-0.5Ag alloy is the best. After being treated by hot extrusion, the mechanical property of the extruded Mg-4Zn-0.5Sr-Ag alloy improved a lot. The improved mechanical property of the extruded Mg-4Zn-0.5Sr-Ag alloy may be caused by dispersion strengthening. As depicted in Figure 7, the coarse second phase in the cast alloy is broken to form a fine second phase after being extruded and evenly distributed in the matrix; the dispersed second phase would be pinned in the grain boundary, which results in a diffusion strengthening effect. Therefore, the extruded Mg-4Zn-0.5Sr-0.5Ag alloy presented excellent mechanical properties. From the pole diagram of EBSD, it can be seen that the plastic deformation of the alloy was mainly conducted along the crystal plane of {0001}, and the maximum pole density is 7.06.

In addition, grain refinement could promote the stability of the alloy in long-term immersion. According to the organization analysis, the as-cast alloy comprised the Mg substrate and coarse second phases along the grain boundary. The corrosion potential of the Mg matrix is lower than that of the second phases, so the galvanic corrosion would occur by forming electric couples between the Mg substrate and coarse second phases when immersed in SBF. Moreover, pores, impurities, and other defects would be generated inescapability in the casted alloy. Hence, electric couples characterized as large cathode–small anode tended to form between the second phases, defects, and matrix. When immersed in SBF, the small anode was corroded expeditiously and, finally, caused serious pitting corrosion. After extrusion treatment, the grain size of the alloy became smaller, and the coarse second phase in the alloy was fragmentized and dispersed in the matrix, which promoted the formation of electric couples characterized as small cathode–large anode between the second phase and Mg matrix. During full immersion, corrosion products covered on the large anode surface could reduce the galvanic corrosion tendency. Therefore, the degradation resistance of the Mg alloy was further increased by extrusion.

## 4. Conclusions

This article presented a systematic testing method for cast and extruded Mg–4Zn–0.5Sr-xAg alloys; the synergistic effect of trace Ag and extruding treatment on the mechanical performance and corrosion properties of the alloys was analyzed. The results show that:The appropriate amount of the Ag element in cast Mg-4Zn-0.5Sr alloy could make the grain of the alloy refined; when the Ag content is 0.5 wt.%, the obtained Mg-4Zn-0.5Sr-0.5Ag alloy has a minimal grain size, that is, 83.28 μm, and the comprehensive properties of the cast Mg-4Zn-0.5Sr-0.5Ag alloy is the best. The average tensile strength (σ_b_), yield strength (σ_s_), elongation (ε), and hardness of the cast Mg-4Zn-0.5Sr-0.5Ag alloy are 168.00 MPa, 88.00 MPa, 12.20%, and 59.90 HV, respectively.After further extruding treatment on the cast Mg-4Zn-0.5Sr-0.5Ag alloy, the grain size of the alloy was significantly refined to 9 μm; at the same time, fine second phases were formed and evenly distributed in the matrix. And then, the mechanical properties of the alloy are significantly enhanced due to the effect of fine crystal strengthening and dispersion strengthening. The σ_b_, σ_s_, λ, and hardness value of the extruded Mg-4Zn-0.5Sr-0.5Ag alloy are 236.00 MPa, 212.00 MPa, 18.97%, and 65.42 HV, respectively.Under the synergistic action of adding the Ag element and extrusion treatment, the grain size of the alloy was significantly refined and the coarse second phase in the alloy became refined to disperse in the matrix, which benefits the formation of electric couples characterized as small cathode–large anode between the second phase and Mg matrix. During full immersion, corrosion products covered on the large anode surface could reduce the galvanic corrosion tendency. Therefore, the degradation resistance of the Mg alloy was further increased by the synergistic action of adding the Ag element and extrusion treatment.

## Figures and Tables

**Figure 1 materials-16-06423-f001:**
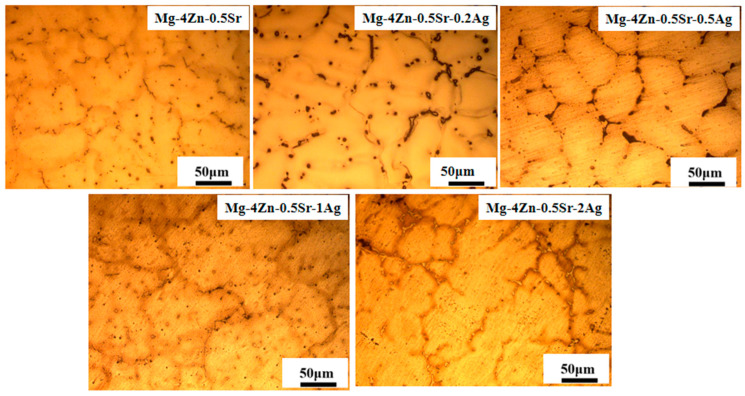
Metallographic structure of casted Mg-4Zn-0.5Sr-xAg (x = 0, 0.2, 0.5, 1, and 2 wt.%) alloys.

**Figure 2 materials-16-06423-f002:**
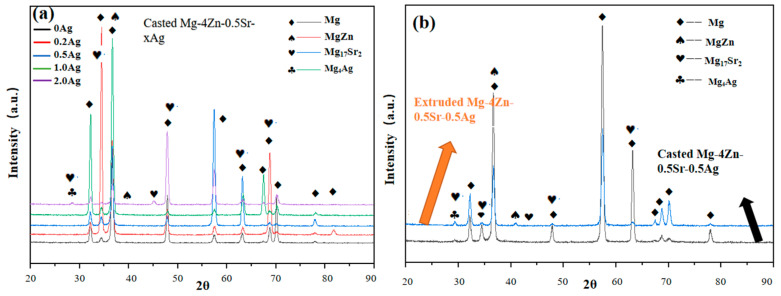
XRD results of Mg alloys: (**a**) is casted Mg-4Zn-0.5Sr-xAg (x = 0, 0.2, 0.5, 1, and 2) alloys, (**b**) is extruded and casted Mg-4Zn-0.5Sr-0.5Ag alloy.

**Figure 3 materials-16-06423-f003:**
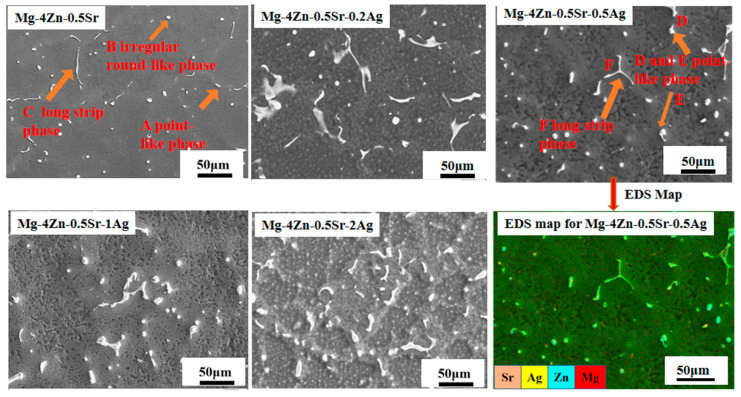
SEM images of casted Mg-4Zn-0.5Sr-xAg (x = 0, 0.2, 0.5, 1, and 2) alloys.

**Figure 4 materials-16-06423-f004:**
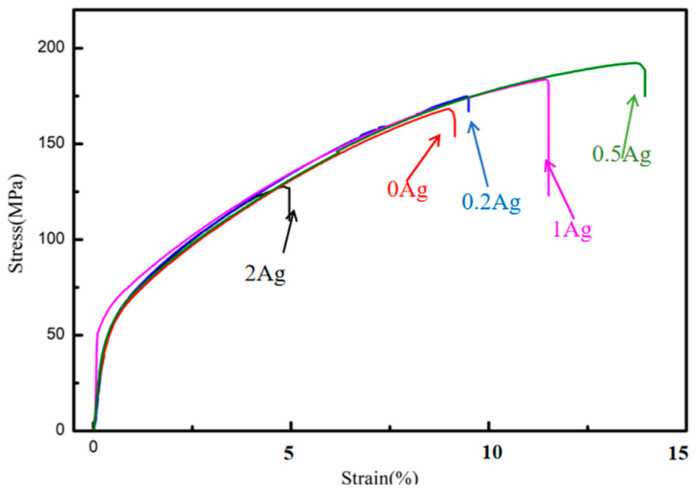
Stress–strain curve of casted Mg-4Zn-0.5Sr-xAg (x = 0, 0.2, 0.5, 1, and 2) alloys.

**Figure 5 materials-16-06423-f005:**
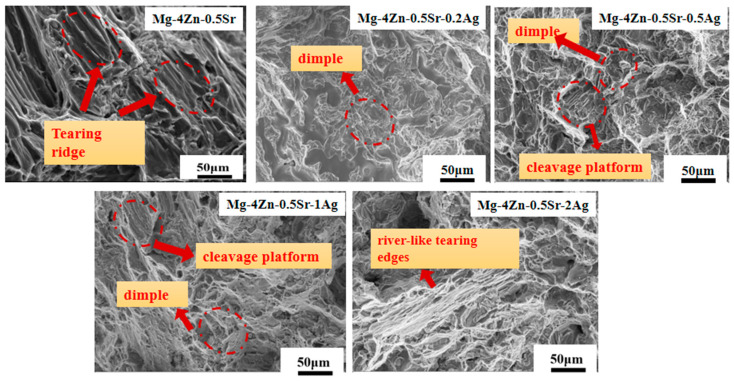
SEM images for tensile fracture surface of cast Mg-4Zn-0.5Sr-xAg (x = 0~2.0 wt.%) alloy.

**Figure 6 materials-16-06423-f006:**
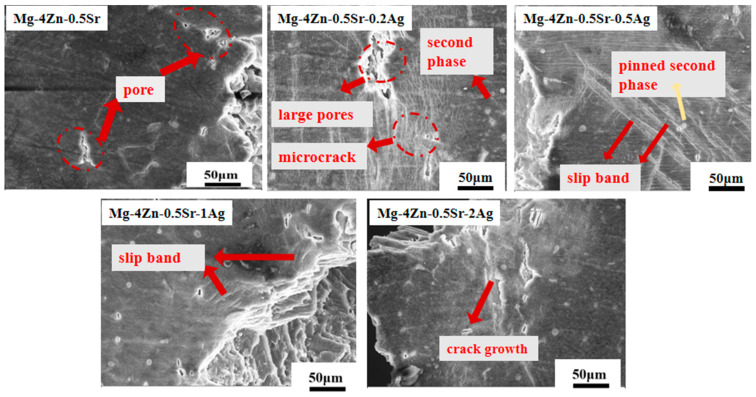
The microstructure near the fracture of cast Mg-4Zn-0.5Sr-xAg (x = 0~2.0 wt.%) alloys.

**Figure 7 materials-16-06423-f007:**
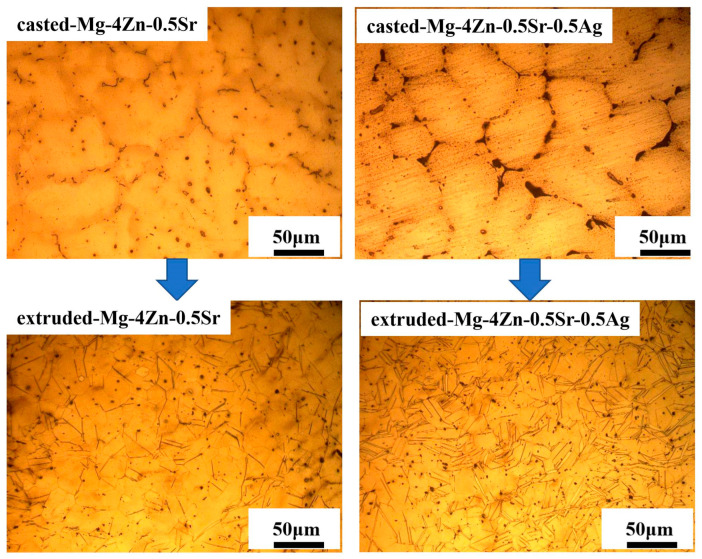
Metallographic morphology for Mg-4Zn-0.5Sr and Mg-4Zn-0.5Sr-0.5Ag alloy before and after extrusion treatment.

**Figure 8 materials-16-06423-f008:**
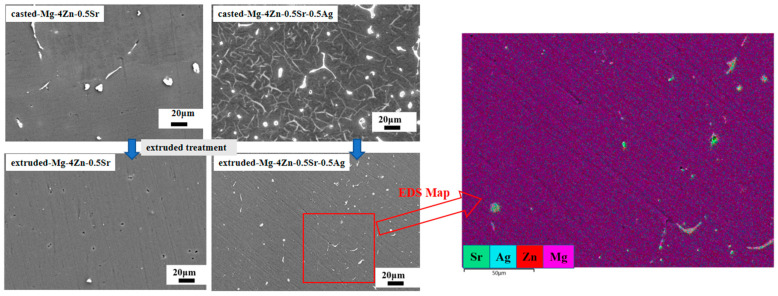
SEM morphology for Mg-4Zn-0.5Sr and Mg-4Zn-0.5Sr-0.5Ag alloy before and after extrusion treatment.

**Figure 9 materials-16-06423-f009:**
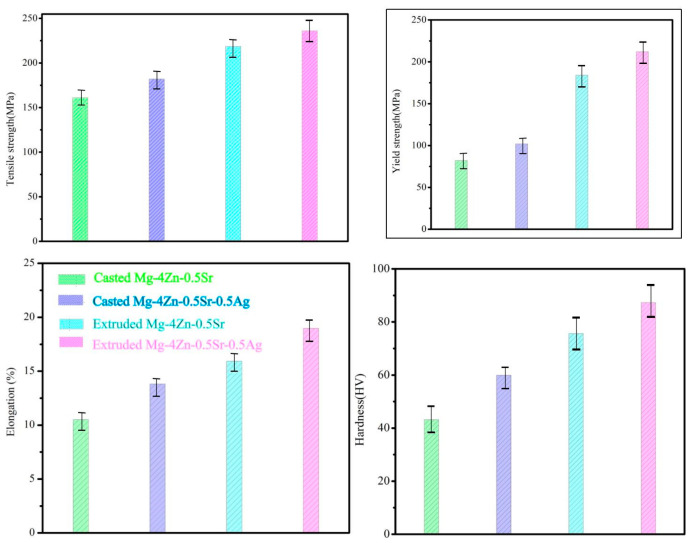
Mechanical properties of casted and extruded Mg-4Zn-0.5Sr and Mg-4Zn-0.5Sr-0.5Ag alloy.

**Figure 10 materials-16-06423-f010:**
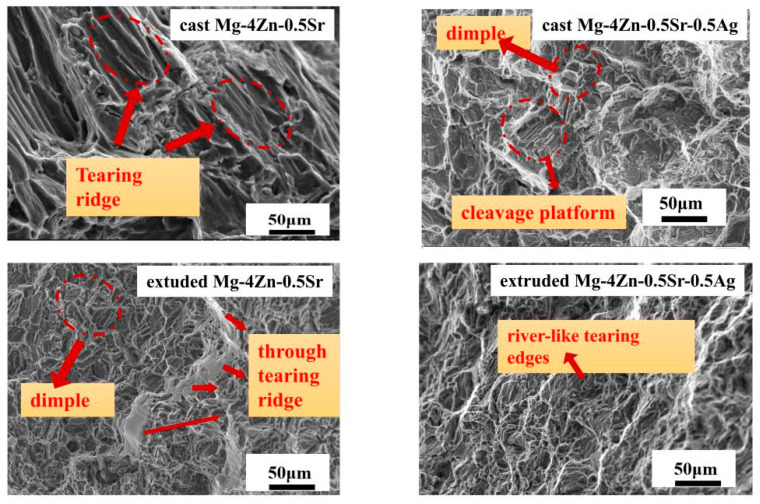
Tensile fracture morphology of casted and extruded Mg-4Zn-0.5Sr and Mg-4Zn-0.5Sr-0.5Ag alloy.

**Figure 11 materials-16-06423-f011:**
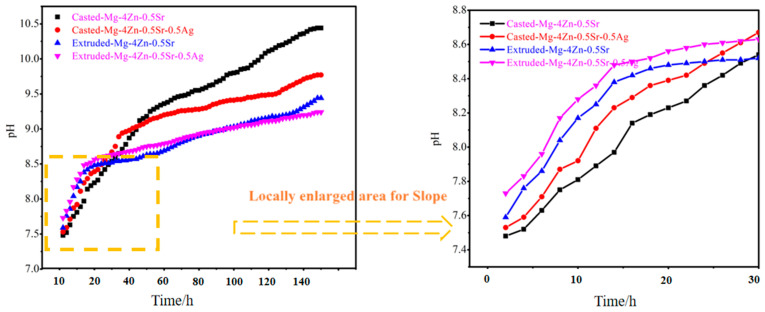
pH value variation tendency of SBF after being immersed with different alloys for different times.

**Figure 12 materials-16-06423-f012:**
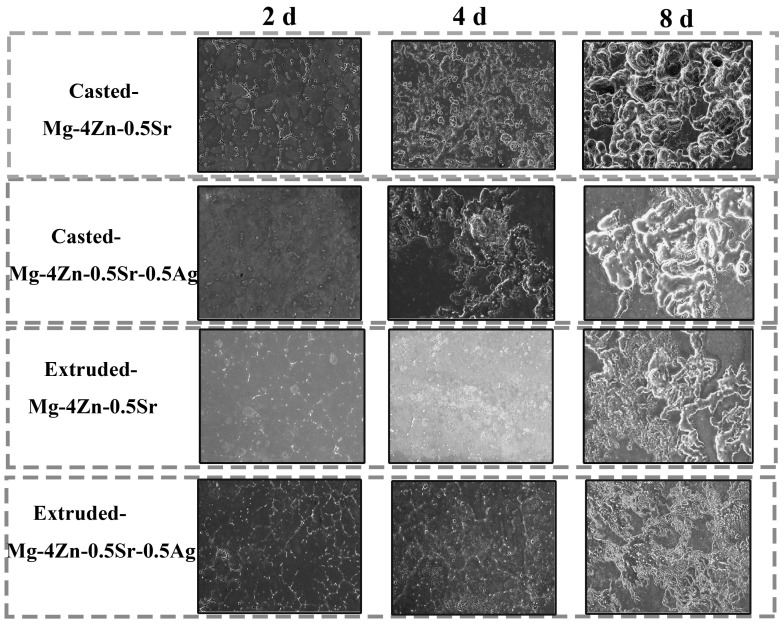
Etching surface of alloys that wipe off corrosion products after dipping in SBF for different times.

**Figure 13 materials-16-06423-f013:**
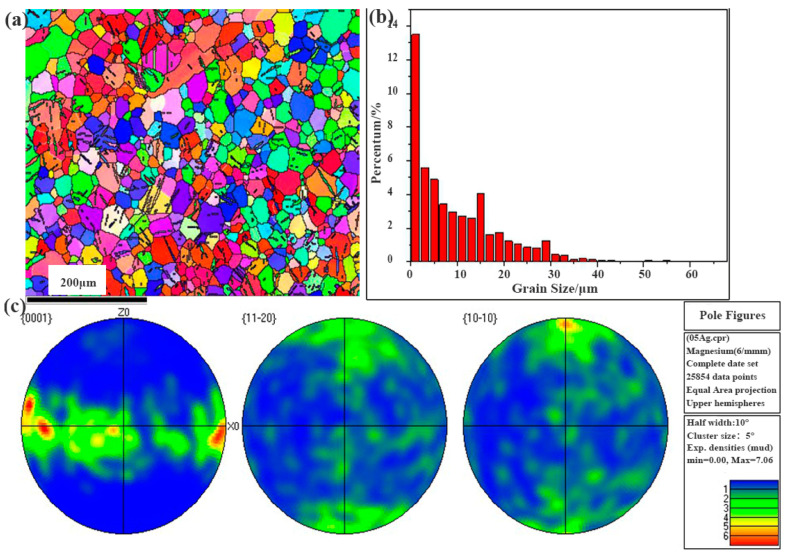
Electron backscatter diffraction (EBSD) of extruded Mg-4Zn-0.5Sr-0.5Ag alloy: (**a**) orientation map of grain; (**b**) grain size distribution; and (**c**) pole figures.

**Table 1 materials-16-06423-t001:** Comparison of theoretical and practical compositions of Mg-4Zn-0.5Sr-xAg alloys.

	Elements (wt.%)	Mg		Zn		Sr		Ag	
Alloys		Theo.	Par.	Theo.	Par.	Theo.	Par.	Theo.	Par.
Mg-4Zn-0.5Sr		95.50	95.69	4.00	3.86	0.50	0.42	-	-
Mg-4Zn-0.5Sr-0.2Ag		95.30	95.00	4.00	4.25	0.50	0.44	0.20	0.18
Mg-4Zn-0.5Sr-0.5Ag		95.00	95.00	4.00	4.18	0.50	0.36	0.50	0.44
Mg-4Zn-0.5Sr-1.0Ag		94.50	94.70	4.00	4.03	0.50	0.34	1.00	0.92
Mg-4Zn-0.5Sr-2.0Ag		93.50	94.30	4.00	3.47	0.50	0.38	2.00	1.82

**Table 2 materials-16-06423-t002:** Grain size for casted and extruded Mg-4Zn-0.5Sr-xAg alloys.

Samples	Casted-MZS	Casted-MZS-0.2Ag	Casted-MZS-0.5Ag	Casted-MZS-1Ag	Casted-MZS-2Ag	Extruded-MZS	Extruded-MZS-0.5Ag
Grain size /μm	103.62	102.67	83.28	102.32	110.43	20	9

**Table 3 materials-16-06423-t003:** EDS analysis of different points in Figure 3 (Point A, B, and C in casted Mg-4Zn-0.5Sr; and Point D, E, and F in Mg-4Zn-0.5Sr-0.5Ag).

	Element	Mg		Zn		Sr		Ag	
Point		wt.%	at%	wt.%	at%	wt.%	at%	wt.%	at%
Mg-4Zn-0.5Sr	A	26.50	49.22	73.50	50.78	-	-	-	-
	B	35.31	59.65	62.90	39.52	1.79	0.84	-	-
	C	36.95	62.06	53.99	33.72	9.06	4.22	-	-
Mg-4Zn-0.5Sr-0.5Ag	D	47.44	72.82	36.87	21.05	8.90	3.79	6.79	2.35
	E	61.29	82.91	21.44	10.79	14.76	5.54	2.51	0.77
	F	43.62	68.74	47.74	27.98	2.60	1.14	6.04	2.14

**Table 4 materials-16-06423-t004:** Mechanical properties of casted Mg-4Zn-0.5Sr-xAg (x = 0, 0.2, 0.5, 1, and 2) alloys.

Alloy	Tensile Strength (σ_b_) (MPa)	Yield Strength (σ_s_) (MPa)	Elongation (ε) (%)	Hardness (HV)
Mg-4Zn-0.5Sr	~161.00	~82.00	~10.30	~43.15
Mg-4Zn-0.5Sr-0.2Ag	~173.00	~97.00	~12.31	~52.90
Mg-4Zn-0.5Sr-0.5Ag	~182.00	~102.00	~13.81	~59.90
Mg-4Zn-0.5Sr-1.0Ag	~175.00	~85.00	~13.12	~51.60
Mg-4Zn-0.5Sr-2.0Ag	~129.00	~70.50	~6.67	~49.50

**Table 5 materials-16-06423-t005:** Corrosion weight loss rate (C_R_) for cast Mg-4Zn-0.5Sr and Mg-4Zn-0.5Sr-0.5Ag, and extruded Mg-4Zn-0.5Sr and Mg-4Zn-0.5Sr-0.5Ag alloy measured after 2 d, 4 d, and 8 d of immersion.

	Immersion Time	2	4	8
Alloys				
Casted Mg-4Zn-0.5Sr		1.05	0.86	0.71
Casted Mg-4Zn-0.5Sr-0.5Ag		0.90	0.68	0.54
Extruded Mg-4Zn-0.5Sr		0.95	0.75	0.67
Extruded Mg-4Zn-0.5Sr-0.5Ag		0.44	0.34	0.25

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
