# Peer review of "The Synergistic Effect of Trace Ag and Hot Extruding on the Microstructure and Properties of a Biodegradable Mg-Zn-Sr-Ag Alloy"

_materials, 2023, doi:10.3390/ma16196423_

Round 1

Reviewer 1 Report

The authors conducted an interesting study. However, there are some questions and remarks:

1) For medical applications, it is important to control gas formation during the decomposition of magnesium alloys, however, the effect on humans is not described in the article.

2) "In our previous work, Mg-4.0 wt% Zn-0.5wt% Sr was best optimized with both me-chanical and degradation behaviors." - Which? links must be provided.

3) What is the accuracy in determining the composition of alloys? what are the dimensions of the ingots?

4) Why are these extrusion parameters chosen?

5) I think it's worth checking for typos. Is there an error in figure 7?

6) How was the grain size determined from Figures 1 and 6? Grain boundaries are not shown. What is the grain size error? Are the Authors sure about the difference in grain size by half after extrusion (Fig. 6)?

7) Can only one peak of the MgZn and Mg17Sr2 phases be noted on the XRD? The description does not explain how and why the peaks changed.

8) It would be nice to present elements EDS-maps for Figure 3.

9) Where does such accuracy in determining mechanical properties come from?

10) Please show the errors in fig. 8.

Author Response

Dear editor,

Thank you very much for your letter and the comments from the reviewers about our

paper submitted to Materials ( materials-2608280). We have checked the manuscript and revised it according to the comments. All changes in the revised manuscript have been highlighted in yellow. If you have any question about this paper, please don’t hesitate to let me know.

Sincerely yours,

Dr. Huishu Wu

College of Mechanical Engineering, Tongling University.

Tel: 0562-5884090

E-mail: ggwuhuishu@163.com

2023/09/13

The authors conducted an interesting study. However, there are some questions and remarks:

Comment 1:

  • For medical applications, it is important to control gas formation during the decomposition of magnesium alloys, however, the effect on humans is not described in the article.

Answer: Thank for your comment. In the paragraph 2 of introduction, the gas formation during the magnesium alloys degradation on the effect of humans was mentioned: Firstly, the hydrogen produced by the fast degradation of the magnesium alloys is released too quickly to be absorbed by the human body, which would cause severe tissue necrosis. Surface technologies of coating, electroless-deposition and physical vapor deposition have been used to delay the degradation process of magnesium alloy[17-18]. Nevertheless, if the inherent corrosion resistance of the magnesium alloy is insufficient, the alloy would still be corroded rapidly when the protective layer of the alloy is damaged. Therefore, developing human friendly Mg based biomaterials with strong inherent corrosion resistance is urgent to meet the clinical application.

Usually, degradation of magnesium alloy in solution is carried out according to the following equation:

Anodic reaction: Mg     Mg2+ +2e- 

Cathodic reaction: H2O+2e-     2OH- + H2

Total reaction: Mg+2H2O     Mg(OH)2+H2  

Based on the degradation process, it can deem that pH value variation for SBF solution and the weight loss rate of the alloy can reflect the hydrogen content, hence, we evaluate the degradation property of magnesium alloys by measured the pH value variation for SBF solution and the weight loss rate of the alloy in this work.

Comment 2:

  • "In our previous work, Mg-4.0 wt% Zn-0.5wt% Sr was best optimized with both mechanical and degradation behaviors." - Which? links must be provided.

Answer: I’m sorry for the inappropriate expression about  “Mg-4.0 wt% Zn-0.5wt% Sr was best optimized”. Related statements have been revised as follows:

 Mg-Zn alloys are promising to develop to be ideal biodegradable implants for bone tissue engineering. Sr element is an important component of human bone, in our previous work[36], Sr was added to improve the biocorrosion and mechanical properties of Mg-4Zn alloy, 0.5wt%Sr is the most appropriate content for corrosion and mechanical properties. However, the property still have room for improvement. Elemnet Ag is a good grain refiner and is kind for human body, by which, Mg-4Zn-0.5Sr-xAg alloys were developed and studied in this work.

[36]J.C. Bian, B.Y. Yu, L. Jiang, J.F. Hao, H.W. Zhu, P.Jin, L.Zheng, R.X. Li, Research on the effect of Sr and Zr on microstructure and properties of Mg-4Zn alloy[J], Int. J. Metalcast., 2021, 4(15):1483-1498.

Comment 3:

  • What is the accuracy in determining the composition of alloys? what are the dimensions of the ingots?

Answer: The accuracy in determining the composition of alloys is mass ratio:wt.%. In this work, the ceramic mold for metal molten is a cylindrical ceramic mold, the outer diameter was 115mm, the inner diameter was 90mm, and the inner wall of the mold was brushed with ZnO painting. The Mg-4Zn-0.5Sr-xAg ingots were machined into cylinders with a diameter of 20mm and length of 65mm. Related descriptions were supplemented in experiment part.

Comment 4:

4) Why are these extrusion parameters chosen?

Answer: The extrusion parameters chosen is based on the type of extruding machine. The type of extruding machine is YH61-500G, the pressure range for the machine is 0-5000KN, the maximum no-load descending speed for the main piston is 0.12m/s, and the slider pressing speed is 0.1-0.6mm/s. The parameters for the extruder has been supplemented into corresponding place.

Comment 5:

5) I think it's worth checking for typos. Is there an error in figure 7?

Answer:Thank for your carefully reading, the caption error (casted-Mg-4Zn-0.5Sr=0.5Ag) in figure has been revised as follows:

Fig.7 Metallographic morphology for Mg-4Zn-0.5Sr and Mg-4Zn-0.5Sr-0.5Ag alloy before and after extrusion treatment

Comment 6:

  • How was the grain size determined from Figures 1 and 6? Grain boundaries are not shown. What is the grain size error? Are the Authors sure about the difference in grain size by half after extrusion (Fig. 6)?

Answer: Grain size was measured by Metallographic method and computed by software of Ipwin32,grain size error is generally 0.5 grain size. For displayed more clearly, the Metallographic image of Mg-4Zn-0.5Sr-0.5Ag was acquired again and replaced, as shown in Fig.1 and Fig.7. Gain size for cast and extruded Mg-4Zn-0.5Sr-xAg alloys is displayed in Table 2. MZS is abbreviation of Mg-4Zn-0.5Sr. The gain size for casted Mg-4Zn-0.5Sr-0.5Ag is 83.28μm, and that is 9 times higher than the grain size of extruded Mg-4Zn-0.5Sr-0.5Ag, which is 9um.  Changes have been marked yellow in article.

Table 2 Gain size for cast and extruded Mg-4Zn-0.5Sr-xAg alloys

Samples

Casted-MZS

Casted-MZS-0.2Ag

Casted-MZS-0.5Ag

Casted-MZS-1Ag

Casted-MZS-2Ag

Extruded-MZS

Extruded-MZS-0.5Ag

Grain size

/μm

103.62

102.67

83.28

102.32

110.43

20

9

Fig.1 Metallographic structure of casted Mg-4Zn-0.5Sr-xAg (x=0,0.2,0.5,1,2wt.%) alloys

Fig.7 Metallographic morphology for Mg-4Zn-0.5Sr and Mg-4Zn-0.5Sr-0.5Ag alloy before and after extrusion treatment

Comment 7

  • Can only one peak of the MgZn and Mg17Sr2phases be noted on the XRD? The description does not explain how and why the peaks changed.

Answer:Thank for your question, we refer to some literature again with this question in mind. We noticed that many peaks aroused by MgZn and Mg17Sr2 phases has been measured[37], so,the XRD patterns have been revised as follows.

Fig.2(a) shows the XRD patterns of Mg-4Zn-0.5Sr-xAg. The results show that for all the samples, the major phases are magnesium(α-Mg) matrix[33], MgZn and Mg17Sr phase[37]. For casted Mg-4Zn-0.5Sr-xAg alloy, peaks from Ag-containing phases is weak, which maybe due to some Ag atoms retain in the solid-solution state, or form precipitates that are too little to be resolved by the XRD experiment[38]. With the Ag content up to 2%, the diffraction peak intensity of Mg4Ag at 28° is increased. At the same time, the intensity change and shift in position of peaks assigned to MgZn and Mg17Sr2 were observed, which is related to the variation in lattice constant of α-Mg caused by the adding of Ag. Then, different degree of solid-solution strengthening effects can be resulted. As shown in Fig.2(b), the intensity of peaks assigned to MgZn (37°) and Mg17Sr2 (34°) is decreased after  extruding treatment. This is because the second phase is dissolved into the matrix during the homogenization process of hot extrusion treatment, and the amount of precipitation is greatly reduced after extrusion (as shown in Fig.7), so the diffraction peak intensity of second phase is significantly reduced.

  • Wang, Y.N. Zhang, P. Hudon,P.Chartrand, I.H. Jung, M. Medra, Experimental study of the crystal structure of the Mg15-xZnxSr3 ternary solid solution in the Mg–Zn–Sr ternary system at 300℃[J],Mater. Des., 2015 86: 305-312.
  • F. Huang,Y.B. Du, W.D. Li, Y.T. Chai, W.G. Huang,Effects of Ag content on the solid-solution and age-hardening behavior of a Mg-5Sn alloy[J], J. Alloy, Compd., 2017, 696: 850-855

Fig.2  XRD patterns of Mg-4Zn-0.5Sr-xAg

Comment 8:

8) It would be nice to present elements EDS-maps for Figure 3.

Answer: We present EDS-maps of casted Mg-4Zn-0.5Sr-0.5Ag in Fig.3 and EDS-maps of extruded Mg-4Zn-0.5Sr-0.5Ag in Fig.8.

 Fig.3 SEM images of casted Mg-4Zn-0.5Sr-xAg(x=0,0.2,0.5,1,2) alloys

Fig.8 SEM morphology for Mg-4Zn-0.5Sr and Mg-4Zn-0.5Sr-0.5Ag alloy before and after extrusion treatment

Comment 9:

9) Where does such accuracy in determining mechanical properties come from?

Answer: The mechanical properties of alloys in Table 3 are average value. Hardness test was conducted on Vickers hardness tester with the load of 0.98 N and a holding time of 30s. Tensile tests conduct on electronic universal testing machine(WGW-100H). Three duplicate samples have been performed to evaluate the mechanical properties of the alloys. So, we add“~” before the datum to indicate these datum are group of average value.

Comment 10:

  • Please show the errors in fig. 8.

Answer: The errors has been added.

Thank you very much for the excellent and professional revision of our manuscript.

Reviewer 2 Report

The article investigates the impact of adding trace amounts of Ag and extrusion treatment on Mg–4Zn–0.5Sr alloys. Results show grain refinement, enhanced mechanical properties, and improved corrosion resistance due to these factors. The article is well-written and presents intriguing results. I have some remarks:

1.     ‘In our previous work, Mg-4.0 wt% Zn-0.5wt% Sr was best optimized with both me- chanical and degradation behaviors ‘. Cite the article.

2.     2.4.2. The chemical reaction is not well written.

3.     Please include the tensile curves from the paper as a new figure to complement Table 3.

4.     Symbol of elongation should be ε not ƛ.

5.     Improve the quality of Fig. 8.

6.     Assign a unique number to each equation in the article and reference them within the text.

7.     Please decrease the size of the legend for Figure 10.

8.     Is it possible to add the EDS Maps in the article?

9.     There are some typographical errors in the manuscript, including instances of double spacing.

Author Response

Dear editor,

Thank you very much for your letter and the comments from the reviewers about our

paper submitted to Materials (Materials-2608280). We have checked the manuscript and revised it according to the comments. All changes in the revised manuscript have been highlighted in yellow. If you have any question about this paper, please don’t hesitate to let me know.

Sincerely yours,

Dr. Huishu Wu

College of Mechanical Engineering, Tongling University.

Tel: 0562-5884090

E-mail: ggwuhuishu@163.com

2023/09/13

The article investigates the impact of adding trace amounts of Ag and extrusion treatment on Mg–4Zn–0.5Sr alloys. Results show grain refinement, enhanced mechanical properties, and improved corrosion resistance due to these factors. The article is well-written and presents intriguing results. I have some remarks:

Comment 1:

  1. ‘In our previous work, Mg-4.0 wt% Zn-0.5wt% Sr was best optimized with both mechanical and degradation behaviors ‘. Cite the article.

Answer: This work has been Cited as Ref[36].

  • C. Bian, B.Y. Yu, L. Jiang, J.F. Hao, H.W. Zhu, P.Jin, L.Zheng, R.X. Li, Research on the effect of Sr and Zr on microstructure and properties of Mg-4Zn alloy[J],  Int. J. Metalcast., 2021, 4(15):1483-1498.

Comment 2:

  1.     2.4.2. The chemical reaction is not well written.

Answer: We are sorry for the problem of format. The written format for chemical reaction has been revised.

Comment 3:

  1. Please include the tensile curves from the paper as a new figure to complement Table 3.

Answer: Tensile curves has been added as shown in Figure 4.

Fig.4  Stress-strain curve of as-cast Mg-4Zn-0.5Sr-xAg alloys(x=0, 0.2, 0.5,1,2 )

Comment 4

  1.     Symbol of elongation should be ε not ƛ.

Answer: The symbol of elongation has been replaced by ε.

Comment 5

  1.     Improve the quality of Fig. 8.

Answer: The revised Fig.8 were shown an follows:

Comment 6:

  1.     Assign a unique number to each equation in the article and reference them within the text.

Answer: Thank for your proposal, the each equation has been assigned number and cited them when state them in text.

Comment 7:

  1.     Please decrease the size of the legend for Figure 10.

Answer: The size for Figure 10 has been decreased.

Comment 8:

  1.     Is it possible to add the EDS Maps in the article?

Answer: The EDS maps for casted and extruded Mg-4Zn-0.5Sr-0.5Ag have been added in Fig.3 and Fig8.

 Fig.3 SEM images of casted Mg-4Zn-0.5Sr-xAg(x=0,0.2,0.5,1,2) alloys

Fig.8 SEM morphology for Mg-4Zn-0.5Sr and Mg-4Zn-0.5Sr-0.5Ag alloy before and after extrusion treatment

Comment 9:

  1.   There are some typographical errors in the manuscript, including instances of double spacing.

Answer: We read through the text carefully and revised typographical errors in the manuscript.
